# Shared Immune and Nutrient Metabolism Pathways Between Autism Spectrum Disorder and Celiac Disease: An In Silico Approach

**DOI:** 10.3390/nu17091439

**Published:** 2025-04-25

**Authors:** Panagiota Sykioti, Panagiotis Zis, Despina Hadjikonstanti, Marios Hadjivassiliou, George D. Vavougios

**Affiliations:** 1Medical School, University of Cyprus, Nicosia 2029, Cyprus; sykpot@hotmail.com (P.S.); vavougyios.georgios@ucy.ac.cy (G.D.V.); 2Sheffield Teaching Hospitals NHS Foundation Trust, Sheffield S10 2JF, UK; m.hadjivassiliou@sheffield.ac.uk

**Keywords:** autism spectrum disorder (ASD), celiac disease (CD), immune dysregulation, nutrient metabolism, gene–disease associations

## Abstract

**Introduction:** Autism spectrum disorder (ASD) is a neurodevelopmental condition characterized by social communication difficulties and repetitive behaviors. Emerging evidence suggests a potential link between ASD and celiac disease (CD), possibly mediated by immune dysregulation and nutrient deficiencies. This study explores the shared biological pathways between ASD and CD using an in silico approach. **Methods:** Gene–disease associations for ASD and CD were retrieved from DisGeNET using MedGen Concept IDs (C1510586 and C0007570, respectively). An over-representation analysis (ORA) was conducted using GeneTrail 3.2 to identify significantly enriched biological pathways, which were then compared for overlap. A false discovery rate (FDR) < 0.05 was considered statistically significant. **Results:** The gene–disease association analysis identified 536 ASD-related genes and 52 CD-related genes. The ORA revealed several shared biological pathways, including immune pathways, cellular metabolism, and micronutrient processing (e.g., folate, selenium, vitamin A). These findings suggest immune dysfunction and nutrient malabsorption as potential mechanistic links between ASD and CD. **Conclusions:** The observed pathway overlap supports the hypothesis that immune dysregulation and metabolic disturbances contribute to both ASD and CD. Nutrient deficiencies, driven by CD-associated malabsorption, may exacerbate ASD symptoms. Additionally, sensory processing abnormalities in ASD could impact dietary choices, complicating gluten-free diet adherence. Future studies should validate these findings in clinical cohorts and explore dietary interventions, such as targeted supplementation, to mitigate ASD symptoms in individuals with CD.

## 1. Introduction

Autism spectrum disorder (ASD) is a complex neurodevelopmental condition characterized by persistent challenges in social communication, alongside restricted, repetitive patterns of behavior, interests, or activities. The term “spectrum” reflects the wide variability in symptoms and severity among individuals. ASD typically manifests in early childhood and can impact functioning across various domains, including social interactions, academic performance, and daily living skills [1]. Although the exact pathophysiology is yet to be determined, it is known that ASD is influenced by both genetic and environmental factors. Genetic studies, including twin and family research, have demonstrated a significant hereditary component, with heritability estimates ranging from approximately 50% to 90% [2]. Recent large-scale genomic studies have highlighted the impact of rare de novo mutations, copy number variations (CNVs), and common polygenic risk factors in ASD susceptibility [3,4]. Notably, disruptions in genes related to synaptic signaling (e.g., SHANK3, NRXN1) and chromatin remodeling (e.g., CHD8, ADNP) have been linked to ASD pathophysiology. Additionally, immune-related genetic variants may contribute to the gut–brain axis dysregulation observed in some autistic individuals, reinforcing the need for further research into gene–environment interactions. Environmental factors also contribute to the ASD risk; prenatal exposures such as maternal infections, air pollution, and epigenetic processes have been associated with an increased likelihood of developing ASD [5]. Additionally, an advanced parental age and perinatal complications have been implicated as potential risk factors [6].

Beyond neurological symptoms, many individuals with ASD also experience a range of gastrointestinal (GI) disturbances, including abdominal pain, constipation, diarrhea, and bloating. Studies suggest that up to 70% of autistic individuals exhibit GI symptoms, a significantly higher prevalence compared to neurotypical individuals [7]. The underlying mechanisms of these GI issues remain unclear, but increasing evidence suggests that an altered gut microbiota, immune dysregulation, and dietary sensitivities could play a role [8]. One of the most debated dietary concerns in ASD is gluten sensitivity, with many families reporting behavioral and GI improvements upon adopting a gluten-free diet (GFD) [9,10].

Celiac disease (CD) is an autoimmune disorder characterized by an abnormal immune response to gluten, leading to chronic inflammation and damage to the small intestine. Celiac disease (CD) manifests in multiple forms, including classical, non-classical, and silent CD, each presenting distinct clinical features. Classical CD is characterized by malabsorption symptoms such as diarrhea and weight loss, while non-classical CD primarily involves extraintestinal manifestations, including neurological and psychiatric symptoms. Silent CD, often asymptomatic, is typically identified through serological and genetic testing [11].

Although ASD and CD are distinct conditions, emerging research suggests a potential overlap in immune-related and genetic pathways. Some studies have identified a higher prevalence of celiac disease-specific antibodies in autistic individuals, even in the absence of classic CD symptoms [12]. This raises important questions about whether gluten-related disorders could contribute to ASD-associated symptoms, particularly in individuals who show clinical improvements on a gluten-free diet.

Despite anecdotal evidence, the scientific community remains divided on the therapeutic efficacy of a gluten-free diet in ASD. Some randomized controlled trials have reported positive effects on behavior, attention, and GI symptoms following gluten-free interventions, while others have found no significant improvements [13,14]. This inconsistency highlights the need for further systematic research to better understand whether certain ASD subgroups may be more responsive to dietary modifications. Investigating the genetic and molecular links between ASD, CD, and gluten sensitivity could provide new insights into personalized therapeutic strategies for autistic individuals.

This study aims to bridge this knowledge gap by leveraging bioinformatics-driven analyses to explore potential shared genetic mechanisms between ASD and CD. Using DisGeNET, a curated gene–disease association database, this study investigates the common genetic pathways that may contribute to immune dysregulation and gut–brain interactions in ASD. Understanding these connections could pave the way for more targeted dietary or immunological interventions for autistic individuals experiencing gluten-related issues.

## 2. Methods

To address our research question, we sought to obtain gene–disease and pathway–disease data for autism spectrum disorder and celiac disease and determine the overlap in gene–disease and variant–disease associations and deregulated pathways. The specific workflow followed to perform these analyses is described below.

**Data Sources:** The data utilized in this study were sourced from DISGENET (available from https://disgenet.com on 2 February 2025), a knowledge management and discovery platform developed to drive inquiries into the genetic foundations of human diseases. The core knowledge assemblies in the DISGENET database structure are the gene–disease association (GDA) and the variant–disease association (VDA), formulated by integrating data from multiple sources described in detail elsewhere [15]. Briefly, these include data from curated datasets on genotype–phenotype associations retrieved from DISGENET’s current version (V24.4), which are supported by more than 1,500,000 publications; these refer to over 6,000,000 GDAs and over 1,000,000 VDAs.

Another important knowledge assembly that is unique to DISGENET is the disease–disease association (DDA). DDAs are used to explore similarities between diseases or between diseases and traits based on shared genes and variants. Analyzing DDAs can help to study disease comorbidities and identify genomic similarities among different diagnoses. DDAs connect two diseases if they share at least one gene or variant from a specific database.

For this analysis, we first identified the entries “autism spectrum disorder” (MedGen Concept ID: C1510586) and “celiac disease” (MedGen Concept ID: C0007570) in DISGENET (accessed on 2 February 2025) and retrieved GDA, VDA, and DDA data for each. Note that, in DISGENET, diseases are identified through MedGen Concept IDs; MedGen is a complimentary database that functions as a portal to information on the genetic aspects of human health and diseases [16].

After retrieving data for each disease, we scrutinized the DDAs to determine whether a disease–disease association was previously identified in the DISGENET data for CD and ASD. Subsequently, we compared the GDAs and VDAs for each to determine such relationships ad hoc (i.e., by identifying overlapping genes and/or variants). Finally, we utilized gene–disease association data to perform pathway-level comparisons, described in the subsequent section.

**Over-representation analysis and pathway-level overlap comparisons:** In the previous step, we searched DisGeNET for the terms “autism spectrum disorder” (MedGen Concept ID: C1510586) and “celiac disease” (MedGen Concept ID: C0007570) and used them to identify their respective gene–disease associations. In step 2, we used the genes in gene–disease associations to construct candidate interactomes for each disease. The rationale for this approach was (a) to determine whether individual GDAs comprise over-represented pathways, forming pathway–disease associations (PDAs), and then (b) to compare those significantly over-represented between ASD and CD in order to detect overlap. Pathway overlap serves to indicate whether ASD and CD are characterized by shared dysregulations in biological mechanisms and is not currently assessed by DisGeNET.

Instead, we leveraged the putative interactomes for ASD and CD to be analyzed by GeneTrail 3.2 (available from http://genetrail.bioinf.uni-sb.de; accessed on 2 February 2025) [17]. The GeneTrail web service is designed to perform enrichment and network analysis procedures, which are crucial computational methods in analyzing high-dimensional datasets and provide novel insights into biological processes. Typically, these approaches employ statistical tests to ascertain whether the biological categories being investigated are deregulated. One of the most widely used techniques in this context is over-representation analysis (ORA), an approach that determines whether a set of genes is significantly more or less present in a biological category than expected by chance. ORA was utilized herein to determine over-represented pathways in ASD and CD based on DisGeNET-extracted interactomes. Subsequently, these pathways could be compared directly to detect overlap, i.e., whether pathways are significantly over-represented in both conditions. To determine significant over-representations, a false discovery rate < 0.05 was applied in order to consider the ORA results statistically significant.

## 3. Results

DisGeNET was accessed on 2 February 2025. The query for “autism spectrum disorder” (MedGen Concept ID: C1510586) and “celiac disease” (MedGen Concept ID: C0007570) retrieved gene–disease association (GDA) data for both ASD (n_genes_ = 536; Gene List/Interactome A) and CD (n_genes_ = 52; Gene List/Interactome B) (available as Appendix A for raw files and candidate interactomes).

Gene Lists A and B were used as input gene lists for over-representation analysis via GeneTrail 3.2. GeneTrail was accessed on 2 February 2025, and the over-representation analysis (ORA) was performed on Lists A and B. This process identified several over-represented biological pathways for each condition. These pathway–disease associations (PDAs) were then compared for overlap between celiac disease and ASD (Appendix A for raw files).

Several biological pathways that are over-represented are shared between ASD and CD. These include, in broad categories, immune pathways, cellular metabolic processes, and nutrient-specific biological pathways such as folate metabolism, the selenium micronutrient network, and vitamin A and carotenoid metabolism. Table 1 presents the results from the Gene Ontology—Biological Pathways database and Table 2 presents the results from WikiPathways.

Figure 1 is a Sankey plot diagram illustrating the association between CD and ASD, taking into consideration their common biological pathways

## 4. Discussion

The present study identified several immune pathways to be over-represented in both ASD and CD. Using a meta-analytical approach, our team has previously estimated that the pooled prevalence of ASD in coeliac disease is 8.7%. Moreover, in this meta-analysis of four studies that included a total of 11,234 patients with CD and 1,042,414 controls, it was established that the odds of having ASD was significantly higher in the CD patients compared to controls (OR 1.53, 95% CI 1.24–1.88, *p* < 0.0001) [18]. This is consistent with emerging studies that increasingly suggest that immune dysfunction is a viable risk factor contributing to the neurodevelopmental deficits observed in ASD [19].

CD and gluten sensitivity (GS) are known to have severe neurological manifestations, often in the absence of gastrointestinal symptoms. The most common neurological manifestations are cerebellar ataxia (gluten ataxia), peripheral neuropathy (gluten neuropathy), and gluten encephalopathy [20]. A strict gluten-free diet (GFD) has been shown to have positive effects on ataxia, peripheral neuropathy, epilepsy, migraines, and cognitive impairments like “brain fog” that have been linked to gluten consumption, and many patients experience significant improvements upon eliminating gluten from their diet. The mechanisms behind these benefits include reduced autoimmune responses, decreased inflammation, and improved nutrient absorption, all of which contribute to better neurological function [20,21,22].

The impact of a strict gluten-free diet (GFD) on neurodevelopmental symptoms varies across different conditions, according to the current literature. In children with ASD, studies have yielded mixed results regarding the efficacy of a GFD. A randomized, controlled, single-blinded trial involving 66 children with ASD found no significant differences in autistic symptoms, maladaptive behaviors, or intellectual abilities between those on a GFD and those on a gluten-containing diet over a six-month period [23]. Conversely, another randomized clinical trial reported that a six-week GFD led to a significant decrease in behavioral disorders and gastrointestinal symptoms in children with ASD [24]. Interestingly, an evaluation of psychological distress in children with CD adhering to a strict GFD indicated that the emotional and behavioral profiles were comparable to those of their healthy peers, suggesting that a GFD may mitigate psychological distress [25]. To our knowledge, there are no dedicated studies of patients with ASD and CD that look into the additional effects of CD on behavior.

Our study, however, may explain the contradictory findings of the role of a gluten-free diet in ASD; it might have an effect only through specific nutrient metabolic pathways that are over-represented in both ASD and CD. As can be seen in Table 2, both ASD and CD share pathways related to folate, selenium, and vitamin A, the absorption of which occurs mainly in the duodenum [26,27,28]. This may be a mechanism through which the two entities, ASD and CD, are linked, as the poor absorption of these nutrients may worsen ASD symptoms, whereas a strict gluten-free diet in patients with CD will improve absorption and lead to improvements in ASD symptoms.

On the other hand, it is known that people with ASD have significant sensory disturbances, affecting multiple sensory modalities. Approximately 90% of individuals with ASD experience atypical sensory responses, which can manifest as hyper-responsiveness (over-reactivity), hypo-responsiveness (under-reactivity), or sensory-seeking behaviors. These sensory abnormalities are now recognized as part of the diagnostic criteria for ASD in the DSM-5 [29]. Research indicates that these sensory issues are pervasive across various sensory domains, including tactile, auditory, visual, and olfactory systems. For instance, individuals with ASD may exhibit heightened sensitivity to touch or sound, leading to discomfort or avoidance behaviors, or they may show reduced sensitivity, resulting in a lack of response to sensory stimuli. These atypical sensory experiences are evident early in development and can significantly impact daily functioning and quality of life [30]. Such sensory deficits are a common manifestation of CD and GS, even in patients who do not have structural pathologies, including peripheral neuropathy [31].

Importantly, sensory disturbances play a critical role in food preferences and dietary choices, especially in children with ASD. Sensory sensitivities often drive children to prefer foods with specific textures, tastes, or appearances, typically favoring processed foods that are bland, crunchy, or visually simple. For example, many children with ASD prefer pasta, bread, chicken nuggets, or dry, salty snacks, like crackers, and are often resistant to trying new foods, particularly those with mixed textures or strong flavors [32]. This selective eating can be problematic, not only as it contributes to nutritional imbalances and challenges in managing a balanced diet but also as, if a comorbidity with CD exists, a gluten-free diet may be completely impossible.

In summary, the pathophysiological mechanisms underlying ASD and CD are multifaceted, encompassing genetic predispositions, environmental influences, immune dysregulation, and nutritional deficiencies (Figure 2). Independent genetic factors may contribute to both conditions, with certain alleles increasing the susceptibility to immune dysfunction, linking indirectly ASD and CD. Environmental factors, such as early-life exposures, can further exacerbate these risks. The immune pathways identified in this study highlight the significant overlap between ASD and CD, suggesting that immune system abnormalities may serve as a common link. Additionally, the shared nutrient metabolic pathways, particularly those involving folate, selenium, and vitamin A, underscore the potential impact of nutrient absorption on the neurodevelopmental outcomes in ASD. The interplay between sensory disturbances in ASD and dietary preferences complicates nutritional management, especially in the context of CD, where dietary restrictions can limit nutrient intake and exacerbate symptoms. This intricate relationship between immune function, nutrition, and sensory processing underscores the need for a comprehensive understanding of these mechanisms in both ASD and CD.

## 5. Limitations

Our findings have some limitations. Firstly, the current research suggests that ASD and CD do not share causal genetic factors [33]. Our scrutiny of DisGeNET similarly suggests that ASD and CD do not share causal single-nucleotide polymorphisms that could account for shared susceptibility, as determined by the lack of shared variant–disease associations. Opting to study the pathway–disease overlap instead acknowledges that pathway overlap between two diseases does not require gene overlap. In this context, our findings reflect how the co-existence of these two conditions may enhance aspects of their detriment, as suggested by a recent meta-analysis of published clinical studies [34]. While relatively simplistic (i.e., utilizing a single resource), our approach re-examines the lack of genetic susceptibility via an alternative concept where the overlap of ASD and CD is characterized by the synergistic dysregulation of specific pathways. As ASD and CD were not directly associated in DisGeNET, and neither gene– or variant–disease associations were discovered, we did not consider utilizing additional databases to avoid false positives, essentially genes that would be a result of fringe rather than salient associations. On the other hand, overlapping deregulated pathways between ASD and CD represent a biologically plausible substrate that can account for any potential synergistic effects. It should be mentioned, however, that only with subsequent validation, such as examining these pathways within these cohorts, would confirmation be possible; this was beyond the means and scope of this study. We do, however, provide the rationale and supporting literature as the foundation for a future validation study to be carried out.

## 6. Conclusions

Looking ahead, future research should focus on elucidating the complex interactions between nutrition and neurodevelopmental disorders, particularly in the context of ASD and CD. Projects aimed at mapping nutrient absorption and metabolism in individuals with ASD, especially those with concurrent CD, will be crucial in identifying potential therapeutic interventions. Investigating the implementation of dietary supplementation with selenium, vitamin A, and folate could provide valuable insights to aid in mitigating ASD symptoms and improving overall health outcomes. Monitoring behavioral changes through validated assessment tools will be essential to evaluate the efficacy of these interventions. By integrating nutritional science with behavioral health, researchers can pave the way for innovative strategies that not only enhance nutrient absorption but also address the sensory and dietary challenges faced by individuals with ASD. This multidisciplinary approach holds promise in improving the quality of life of those affected by these conditions and may lead to more tailored and effective treatment protocols.

## Figures and Tables

**Figure 1 nutrients-17-01439-f001:**
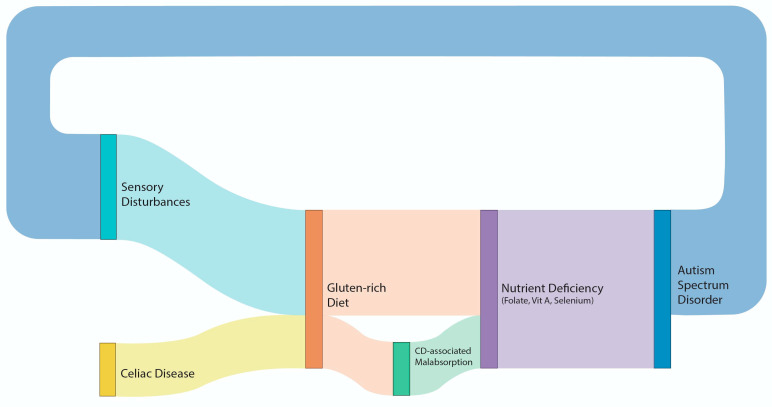
Sankey plot diagram illustrating the association between CD and ASD, taking into consideration their common biological pathways.

**Figure 2 nutrients-17-01439-f002:**
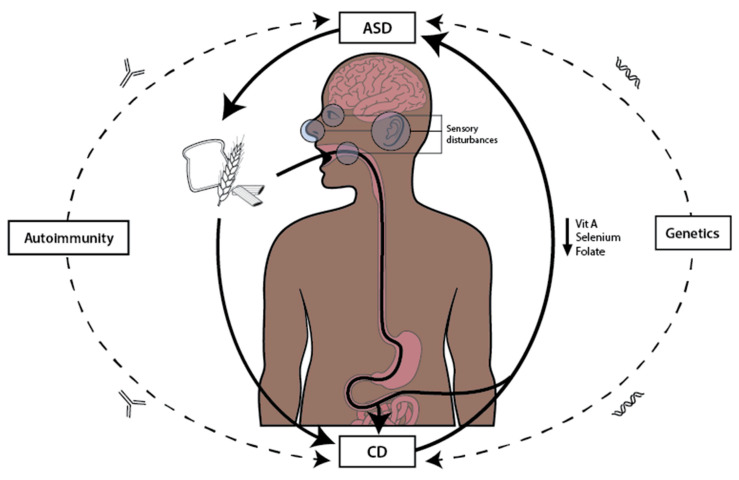
The figure illustrates the vicious cycle between autism spectrum disorder (ASD), celiac disease (CD), and nutrition. ASD influences specific dietary choices, often characterized by selective eating and gluten-containing foods, which contributes to the worsening of CD. This exacerbation leads to impaired gut integrity and the malabsorption of key nutrients, such as vitamin A, selenium, and folate, essential for neurological function. Nutrient deficiencies, in turn, aggravate ASD symptoms, perpetuating the cycle. Additionally, genetic predisposition and autoimmune mechanisms serve as additional links between ASD and CD.

**Table 1 nutrients-17-01439-t001:** Over-represented gene ontologies from the Gene Ontology (GO) database that are shared between both autism spectrum disorder and celiac disease. Significant over-representation is determined by a false discovery rate of less than 0.05. The significantly over-represented biological processes presented herein can be broadly categorized as cellular homeostatic/housekeeping processes, immune pathways, and pathways associated with the response to nutrients. ASD: autism spectrum disorder; CD: celiac disease; FDR: false discovery rate.

Gene Ontology	FDR_ASD_	FDR_CD_
**Response to Stimulus**	1.37 × 10^−5^	3.78 × 10^−12^
**Regulation of Cell Population Proliferation**	6.47 × 10^−5^	3.66 × 10^−9^
**Response to Lipopolysaccharide**	0.0189	4.09 × 10^−8^
**Response to Molecules of Bacterial Origin**	0.0324	6.69 × 10^−8^
**Response to Lipids**	4.82 × 10^−7^	9.71 × 10^−8^
**Response to Organic Substances**	3.22 × 10^−13^	6.90 × 10^−7^
**Positive Regulation of Cell Population Proliferation**	0.0025	4.62 × 10^−6^
**Positive Regulation of Cell Differentiation**	0.0079	8.70 × 10^−6^
**Positive Regulation of Transport**	0.0482	6.58 × 10^−5^
**Cellular Response to Organic Substances**	1.01 × 10^−7^	1.01 × 10^−4^
**Regulation of Cell Differentiation**	5.51 × 10^−4^	1.02 × 10^−4^
**Chemical Homeostasis**	2.93 × 10^−7^	1.94 × 10^−4^
**Homeostatic Process**	1.98 × 10^−8^	2.75 × 10^−4^
**Positive Regulation of Signal Transduction**	0.0076	6.97 × 10^−4^
**Positive Regulation of Cation Transmembrane Transport**	0.0482	0.0015
**Positive Regulation of Ion Transmembrane Transport**	0.0125	0.0021
**Cellular Homeostasis**	1.26 × 10^−5^	0.0025
**Cellular Process**	7.01 × 10^−18^	0.0026
**Cellular Metal ion Homeostasis**	0.0131	0.0031
**Cellular Chemical Homeostasis**	4.00 × 10^−4^	0.0035
**Cell Motility**	1.26 × 10^−5^	0.0042
**Retinoid Metabolic Process**	2.35 × 10^−5^	0.0049
**Metal Ion Homeostasis**	2.96 × 10^−4^	0.0055
**Diterpenoid Metabolic Process**	1.04 × 10^−5^	0.0056
**Cellular Cation Homeostasis**	0.0017	0.0060
**Cellular Ion Homeostasis**	0.0025	0.0068
**Biological Process**	2.16 × 10^−15^	0.0068
**Response to Amyloid-Beta**	0.0083	0.0071
**Locomotion**	1.04 × 10^−5^	0.0078
**Cell Migration**	0.0021	0.0081
**Response to Oxidative Stress**	0.0303	0.0085
**Catabolic Process**	9.14 × 10^−8^	0.0089
**Response to Inorganic Substances**	0.0011	0.0094
**Positive Regulation of Metabolic Process**	1.97 × 10^−5^	0.0099
**Cation Homeostasis**	1.61 × 10^−5^	0.0104
**Isoprenoid Metabolic Process**	2.86 × 10^−5^	0.0105
**Positive Regulation of Nitrogen Compound Metabolic Process**	0.0039	0.0150
**Positive Regulation of Ion Transport**	0.0209	0.0164
**Ion Homeostasis**	1.03 × 10^−4^	0.0190
**Regulation of Signalling**	4.12 × 10^−4^	0.0191
**Regulation of Cellular Component Organization**	0.0010	0.0214
**Positive Regulation of Cellular Metabolic Process**	0.0020	0.0220
**Response to Nutrient Levels**	3.32 × 10^−4^	0.0229
**Cellular Response to Drugs**	0.0067	0.0243
**Positive Regulation of Cellular Biosynthetic Process**	0.0278	0.0249
**Response to Vitamins**	2.44 × 10^−4^	0.0255
**Positive Regulation of Gene Expression**	0.0010	0.0257
**Alcohol Metabolic Process**	2.07 × 10^−9^	0.0258
**Carboxylic Acid Metabolic Process**	5.93 × 10^−18^	0.0287
**Response to Extracellular Stimulus**	2.92 × 10^−5^	0.0287
**Cell Communication**	1.94 × 10^−7^	0.0290
**Monocarboxylic Acid Metabolic Process**	8.55 × 10^−18^	0.0358
**Response to Nutrient**	1.27 × 10^−6^	0.0379
**Cellular Lipid Metabolic Process**	4.87 × 10^−10^	0.0416

**Table 2 nutrients-17-01439-t002:** Over-represented pathways from the WikiPathways database that are shared between both autism spectrum disorder and celiac disease. Significant over-representation is determined by a false discovery rate of less than 0.05. The significantly over-represented pathways presented herein can be broadly categorized to nutrient metabolism and gene expression regulation. ASD: autism spectrum disorder; CD: celiac disease; FDR: false discovery rate.

Gene Ontology	FDRA_SD_	FDR_CD_
**Folate Metabolism**	0.0074	6.37 × 10^−4^
**Selenium Micronutrient Network**	9.64 × 10^−4^	0.0013
**Vitamin A and Carotenoid Metabolism**	0.0420	0.0024
**Nuclear Receptors Meta-Pathway**	9.43 × 10^−9^	0.0121
**Aryl Hydrocarbon Receptor Pathway**	3.11 × 10^−5^	0.0393
**Genes Involved in Male Infertility**	4.12 × 10^−5^	0.0393

## Data Availability

Data were provided as Appendix A and will be also available upon reasonable request.

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
