# Peer review of "Shared Immune and Nutrient Metabolism Pathways Between Autism Spectrum Disorder and Celiac Disease: An In Silico Approach"

_nutrients, 2025, doi:10.3390/nu17091439_

Round 1

Reviewer 1 Report

Comments and Suggestions for Authors

It is an important study to find the relationship between ASD and CD. However, a few concerns need to be addressed such as.,

-> Authors are not clear about how many patients data they have analyzed in each ASD and CD category? Does the overlap show a statistical significance? include the statistical data as well in the manuscript with appropriate descriptions.

-> Also, the authors have mentioned in the first paragraph of the discussion that “ASD was significantly higher in the CD patients compared to controls” , in that case how many control subjects were included in their analysis?

-> In the method section if the authors can describe more about the DisGeNet and GeneTrial 3.2 methods including its prior usage in other disease models or in different aspect of ASD itself, will only strengthen the methodology in this study by also providing more references.

-> Introduction is very brief. Describe more about ASD, its connection with gluten and Celiac Disease with appropriate references. Then describe the importance of your study.  

-> Authors have not provided the supplementary material in the submission as they have mentioned in the results. 

Author Response

Comment 1. Authors are not clear about how many patients data they have analyzed in each ASD and CD category? Does the overlap show a statistical significance? include the statistical data as well in the manuscript with appropriate descriptions.

Response: Thank you for your comment. In this study, we did not perform an analysis on patient data. We aimed to determine whether ASD and CD shared genetic dysregulations by examining disesase-disease, gene-disease, and variant-disease associations via DisGeNET; furthermore, we performed over-representation analysis utilizing gene-disease association data to identify shared deregulated pathways, and had not been previously performed when comparing ASD and CD’s pathophysiology.

Comment 2. Also, the authors have mentioned in the first paragraph of the discussion that “ASD was significantly higher in the CD patients compared to controls” , in that case how many control subjects were included in their analysis?

Response: This refers to a previous meta-analysis of 4 studies that included a total of 11,234 patients with CD and 1,042,414 controls – we included this in the sentence for clarity

Comment 3. In the method section if the authors can describe more about the DisGeNet and GeneTrial 3.2 methods including its prior usage in other disease models or in different aspect of ASD itself, will only strengthen the methodology in this study by also providing more references.

Response. Thank you for your comment and excellent suggestion. Per your guidance, we have significantly expanded the methods section to expand on the rationale, tools, procedures and concepts utilized in this study. We have also included appropriate references that present the DisGeNET and GeneTrail and their central role in omics research.

Comment 4. Introduction is very brief. Describe more about ASD, its connection with gluten and Celiac Disease with appropriate references. Then describe the importance of your study.  

Response. We expanded our introduction as per your guidance and described the importance of this study more.

Comment 5. Authors have not provided the supplementary material in the submission as they have mentioned in the results. 

Response. Thank you for your comment. We have provided the necessary information in-text and as supplementary materials 1 and 2.

Reviewer 2 Report

Comments and Suggestions for Authors

Summary
The manuscript should be formatted according to MDPI’s template, and the line numbers should be included. Several key points need to be addressed in the Discussion section. Additionally, the Results section should be strengthened, and the Materials and Methods (M&M) section should be more detailed. The Conclusion must be more focused.

Title
Ensure that the title adheres to MDPI’s Instructions for Authors by capitalizing each relevant word.

Introduction
The authors should provide an overview of the different types of celiac disease (https://doi.org/10.1053/j.gastro.2008.09.069) and their potential genetic relationships. Additionally, they should discuss the implications of genetic variants in autism spectrum disorder (ASD). Within this context, I recommend incorporating the following relevant references: https://doi.org/10.1038/s41598-024-66475-2; https://doi.org/10.1038/s41591-023-02408-2

Materials and Methods
The manuscript should include a dedicated M&M section, incorporating details currently presented in the Results section, such as:

"DisGeNET was accessed on 02/02/2025. The query for 'Autism Spectrum Disorder' (MedGen Concept ID: C1510586) and 'Celiac Disease' (MedGen Concept ID: C0007570) retrieved Gene-Disease Association (GDA) data for both ASD (n_genes = 536; Gene list A) and CD (n_genes = 52; Gene list B) (available as Supplementary Material).
Gene List A and B were used as input gene lists for over-representation analysis via GeneTrail 3.2. GeneTrail was accessed on 02/02/2025, and Over Representation Analysis (ORA) was performed on lists A and B. This process identified several over-represented biological pathways for each condition. These Pathway-Disease Associations (PDAs) were then compared for overlap between Celiac Disease and ASD."

Additionally, I encourage the authors to explore further databases such as Malacards or OMIM and discuss the genes associated with susceptibility to celiac disease.

Results
The authors should provide a more detailed description of Table 1.
Additionally, I encourage incorporating graphical representations, such as Sankey plot diagrams, to better illustrate the association between CD and ASD and their common associated genes.

Discussion
The authors should elaborate on the statement: "It has been demonstrated that a strict gluten-free diet has positive effects in all forms of neurological manifestations."
According to the literature, are there notable differences in neurodevelopmental symptoms between patients following a strict gluten-free diet and those who do not? Given that many celiac cases remain undiagnosed, this aspect should be discussed in relation to potential neurological manifestations.
Furthermore, the genetic features should be explored in greater depth.

Conclusion
The authors should justify the statement: "Our scrutiny of DisGeNET similarly suggests that ASD and CD do not share causal single nucleotide polymorphisms that could account for shared susceptibility."
In fact, specific polymorphisms or gene dysregulation have not been explicitly mentioned in the manuscript and should be discussed accordingly.

Author Response

Comment 1. The manuscript should be formatted according to MDPI’s template, and the line numbers should be included. Several key points need to be addressed in the Discussion section. Additionally, the Results section should be strengthened, and the Materials and Methods (M&M) section should be more detailed. The Conclusion must be more focused.

Response. We added the line numbers and expanded significantly all sections of the manuscript.

Comment 2. Title: Ensure that the title adheres to MDPI’s Instructions for Authors by capitalizing each relevant word.

Response. We capitalised the relevant words – if any more editing is needed please feel free to make the changes before proof reading.

Comment 3. Introduction: The authors should provide an overview of the different types of celiac disease (https://doi.org/10.1053/j.gastro.2008.09.069) and their potential genetic relationships. Additionally, they should discuss the implications of genetic variants in autism spectrum disorder (ASD). Within this context, I recommend incorporating the following relevant references: https://doi.org/10.1038/s41598-024-66475-2; https://doi.org/10.1038/s41591-023-02408-2

Response. We made the suggested changes and incorporated the comments of Reviewer 1 too in an expanded introduction. We have also included all these references which were appropriate.

Comment 4. Materials and Methods: The manuscript should include a dedicated M&M section, incorporating details currently presented in the Results section, such as:

"DisGeNET was accessed on 02/02/2025. The query for 'Autism Spectrum Disorder' (MedGen Concept ID: C1510586) and 'Celiac Disease' (MedGen Concept ID: C0007570) retrieved Gene-Disease Association (GDA) data for both ASD (n_genes = 536; Gene list A) and CD (n_genes = 52; Gene list B) (available as Supplementary Material).
Gene List A and B were used as input gene lists for over-representation analysis via GeneTrail 3.2. GeneTrail was accessed on 02/02/2025, and Over Representation Analysis (ORA) was performed on lists A and B. This process identified several over-represented biological pathways for each condition. These Pathway-Disease Associations (PDAs) were then compared for overlap between Celiac Disease and ASD."

Additionally, I encourage the authors to explore further databases such as Malacards or OMIM and discuss the genes associated with susceptibility to celiac disease.

Response. Thank you for your comment and excellent suggestion. Per your guidance, we have significantly expanded and rewritten the methods section to expand on the rationale, tools, procedures and concepts utilized in this study. We have also included appropriate references that present the DisGeNET and GeneTrail and their central role in omics research. As you have suggested, appropriate sections from the results have been re-incorporated in the Methods section.

We acknowledge the reviewer’s suggestion on Malacards and OMIM and added a relevant section in “Limitations”. Considering that we did not identify salient disease-disease (ASD – CD), gene-disease and variant-disease associations in the DisGeNET dataset, we did not consider adding additional databases due to potential overlap of primary data sources (e.g., textmining and other curated datasets). Furthermore, as DisGeNET is specifically designed to address the concept of disease-disease genomic overlap in a streamlined manner, aside from potential source overlap, adding additional databases would require introducing de novo workflows and thus deviate from a single workflow to compare GDAs and VDAs between ASD and CD.

Comment 5. Results: The authors should provide a more detailed description of Table 1.
Additionally, I encourage incorporating graphical representations, such as Sankey plot diagrams, to better illustrate the association between CD and ASD and their common associated genes.

Response. We changed the description of Table 1. We already had  an illustration about the different links between the two conditions (CD and ASD) but we also added a Sankey plot diagram as suggested (modified as it had to have a loop). However, it must be noted that our studies did not identify common associated genes but common biological pathways

Comment 6. Discussion: The authors should elaborate on the statement: "It has been demonstrated that a strict gluten-free diet has positive effects in all forms of neurological manifestations."
According to the literature, are there notable differences in neurodevelopmental symptoms between patients following a strict gluten-free diet and those who do not? Given that many celiac cases remain undiagnosed, this aspect should be discussed in relation to potential neurological manifestations.Furthermore, the genetic features should be explored in greater depth.

Response. We have elaborated on the highlighted statement and discussed the differences in neurodevelopmental symptoms following a GFD and the existing controversy on this. As mentioned above our study did not identify common gene-disease or variant-disease associations but common biological deregulated pathways. The specific pathways involved indicate nutrient metabolism as an overarching array of processes plays an important role for both and can be potentially modulated by dietary interventions.

Comment 7. Conclusion: The authors should justify the statement: "Our scrutiny of DisGeNET similarly suggests that ASD and CD do not share causal single nucleotide polymorphisms that could account for shared susceptibility."
In fact, specific polymorphisms or gene dysregulation have not been explicitly mentioned in the manuscript and should be discussed accordingly.

Response. Thank you for your comment. We have clarified the context and interpretations applicable to our findings by the expanded Methods section and the appropriate section in-text. Briefly, over-representation analysis serves to identify deregulated pathways associated with each diseases; whereas variant-disease associations (VDAs; supplementary materials 1) serve to indicate said associations between each disease and specific polymorphisms. Neither a direct disease-disease association (DDA) was identified, nor an indirect overlap between causative variants in our scrutiny of the DisGeNET data.

Round 2

Reviewer 1 Report

Comments and Suggestions for Authors

Authors are advised to mark the specific corrections and additions in a different color in their revision and resend the article. Right now, in this revision there is no markings of such corrections at all. So, I can't check the corrections. Also, the supplementary data zip file "MACOSX" can't be opened, please zip it rename and resend (or make it viewable).

Author Response

Thank you for your comment - the tracked changes version was available already - we are re-uploading it. We kindly ask the Editorial Manager to forward this version to reviewer #1 separately too.

We have renamed the zip file - please also send each file to reviewer #1 separately to make sure that they are openable. 

Reviewer 2 Report

Comments and Suggestions for Authors

Authors addressed all the reviewer's comments

Author Response

Thank you for the acceptance

Round 3

Reviewer 1 Report

Comments and Suggestions for Authors

The authors have improved their manuscript on revision.